# Organotin (IV) Dithiocarbamate Compounds as Anticancer Agents: A Review of Syntheses and Cytotoxicity Studies

**DOI:** 10.3390/molecules28155841

**Published:** 2023-08-03

**Authors:** Nurul Amalina Abd Aziz, Normah Awang, Kok Meng Chan, Nurul Farahana Kamaludin, Nur Najmi Mohamad Anuar

**Affiliations:** Center for Toxicology and Health Risk Studies, Faculty of Health Sciences, Universiti Kebangsaan Malaysia, Jalan Raja Muda Abdul Aziz, Kuala Lumpur 50300, Malaysia; p119419@siswa.ukm.edu.my (N.A.A.A.); chan@ukm.edu.my (K.M.C.); nurulfarahana@ukm.edu.my (N.F.K.); nurnajmi@ukm.edu.my (N.N.M.A.)

**Keywords:** organotin (IV), dithiocarbamate, synthesis, characterization, cytotoxicity

## Abstract

Organotin (IV) dithiocarbamate has recently received attention as a therapeutic agent among organotin (IV) compounds. The individual properties of the organotin (IV) and dithiocarbamate moieties in the hybrid complex form a synergy of action that stimulates increased biological activity. Organotin (IV) components have been shown to play a crucial role in cytotoxicity. The biological effects of organotin compounds are believed to be influenced by the number of Sn-C bonds and the number and nature of alkyl or aryl substituents within the organotin structure. Ligands target and react with molecules while preventing unwanted changes in the biomolecules. Organotin (IV) dithiocarbamate compounds have also been shown to have a broad range of cellular, biochemical, and molecular effects, with their toxicity largely determined by their structure. Continuing the investigation of the cytotoxicity of organotin (IV) dithiocarbamates, this mini-review delves into the appropriate method for synthesis and discusses the elemental and spectroscopic analyses and potential cytotoxic effects of these compounds from articles published since 2010.

## 1. Introduction

Cancer is a disease characterized by aberrant signaling and metabolism resulting in the unchecked proliferation and persistence of mutated cells [1]. Typically, this is driven by the activation of oncogenes or deactivation of tumor suppressor genes, which ultimately disrupts the regulation of the cell cycle and the inhibition of apoptotic pathways [2]. Chemotherapy, surgery, and radiotherapy are commonly used to treat cancer. Chemotherapeutic medications can eradicate cancer cells or regulate their growth throughout the body, whereas radiation or surgery are restricted to localized areas [3]. The effectiveness of chemotherapy lies in its ability to leverage genotoxicity, a process by which chemotherapeutic drugs selectively target cancer cells and produce reactive oxygen species that effectively decimate cancer cells. Chemotherapy is widely acknowledged as the most efficacious treatment option for cancer, either as a standalone approach or in conjunction with radiotherapy [4].

Metal-based cancer chemotherapies have attracted significant attention since cisplatin was first introduced [5,6]. This may be due to the unique properties of metals, such as their ability to undergo redox reactions, varied coordination modes, and reactivities towards organic substances. In addition, various metal-based compounds have been synthesized by modifying and replacing ligands with chemical structures, many of which have shown improved effectiveness in killing cancer cells and improved pharmacokinetic properties [7]. Metal ions and non-labile coligands form covalent bonds with crucial biomolecules such as DNA, proteins, and enzymes. This binding hinders their functionality and causes cell death via various cellular pathways such as apoptosis and necrosis [8]. Upon entering the cell, cisplatin unleashes its cytotoxic properties through the detachment of a chloride ligand, binding to DNA to create intra-strand DNA adducts, and impeding both DNA synthesis and cellular proliferation [9]. Platinum-based cancer therapies have become widely accepted as the norm for treating a diverse range of malignancies, such as ovarian, cervical, lung, head and neck, bladder, and lymphoma cancers [10,11]. However, adverse effects and acquired resistance to platinum-based cancer therapies have been reported [12,13], leading to increased research efforts in the search for new platinum-based treatments.

Organotins have emerged as a promising class of biologically active metal-based therapeutic agents. Although the toxic side effects of organotin (IV) compounds are concerning, new ways to reduce these toxic side effects and improve compound properties are available [14]. Organotin (IV) compounds have been shown to be effective biological agents with antibacterial, antimalarial, schizonticidal, and anticancer activities [15,16,17,18,19]. Their high catalytic and redox capacities, structural versatility, ligand exchange potential, and broad potential contact with biologically beneficial properties make organotin (IV) compounds attractive for a wide variety of applications [14,20,21]. The biological activities of organotin compounds are generally expressed in the following order: RSnX_3_ < R_2_SnX_2_ < R_4_Sn << R_3_SnX, with tri-organotin (IV) substituents possessing the greatest impacts [22,23,24,25,26,27]. Anionic X groups, such as chloride, fluoride, oxide, hydroxy, carboxylate, and thiolate [28], have been reported to have very little effect on their activities [25,27,29]. However, combining two biologically active moieties in the same molecule can enhance its activity [27,30]. The biological effects of organotin compounds can be influenced by the presence of one or more C-Sn bonds, depending on the number and type of alkyl (R) substituents connected to the Sn center [20,31]. The longer the alkyl chain on an organotin compound, the lower its toxicity [25,29]; however, toxicity can also vary depending on the test organism [25]. In addition, compounds containing aryl groups are less toxic than those containing alkyl groups [32,33].

According to Pfeffer et al. [34], apoptosis, a naturally occurring process of cell death, holds immense promise for the development of effective anticancer therapies. Tin complexes have been suggested as potential therapeutic substitutes for cisplatin due to their similar chemical characteristics [35,36,37,38,39]. Although their precise mode of action is unclear, organotin compounds bear a striking resemblance in the mechanism of action to compounds such as cisplatin [23,40] which are believed to bind to the external phosphate groups of DNA and interfere with internal phospholipid metabolism [23,41], thereby causing apoptotic cell death. Hence, the possibility of using organotin compounds as a viable option for developing anticancer medications cannot be dismissed. Moreover, organotin derivatives have shown great potential as therapeutic agents in a variety of tumor cells [20,42], including those associated with ovarian, lung, kidney, colon, prostate, and breast cancers, and melanoma [43,44,45]. They have also shown significant selectivity for several cancer cell lines despite the diversity of their ligands [23,41,46,47].

Dithiocarbamates, including organotin (IV), are a class of dithiocarbamate metals that have been extensively investigated for their beneficial biological potential [44,48,49]. These complexes have rare stereoelectronic properties caused by the sulfur atoms of the dithiocarbamate ligand, which assist in the transport of molecules to target sites and prolong their retention time [48], making them widely used in medicinal chemistry [50,51,52]. The organotin (IV) dithiocarbamate compound has recently attracted attention as a chemotherapeutic agent because of its ability to stabilize a specific stereochemistry and its good anti-proliferative activity, as noted in in vitro studies [44]. Organotin (IV) dithiocarbamate compounds have shown a cytotoxic effect on various cancer cell lines [37,44,46,53], whereas tri-organotin complexes have yielded the highest toxicity effects [37,44,54,55,56].

Mamba et al., Menezes et al., and Syed Annuar et al. claimed that the individual properties of the organotin (IV) and dithiocarbamate constituents may have synergistic effects that enhance biological activities [23,57,58]. Metal dithiocarbamate complexes are insoluble in water yet highly soluble in organic solvents. The chelation of Sn ions by dithiocarbamate ligands reduces metal ion polarity, promotes lipophilicity, and increases permeability, resulting in the increased biological activity of organotin (IV) dithiocarbamate compounds [17,54,59]. The two sulfur atoms present in the molecule provide the dithiocarbamate ligands with strong metal-binding abilities. They can act as inhibitors of enzyme responsible for cancer growth (such as catalase), altering the production of reactive oxygen species, or triggering the induction of apoptosis in the mitochondria [60]. Consequently, in this review, we summarized a specific methodology for synthesizing organotin (IV) dithiocarbamate compounds and their characterization using elemental and spectroscopic analyses. In addition, we investigated their cytotoxic potential against various cancer cells.

## 2. Background Chemistry of Organotin (IV) Dithiocarbamate

Tin, a group 14 post-transition metal, exists in two main oxidation states: tin (II) and tin (IV) [14]. Nevertheless, most of the known tin compounds are organotin (IV) derivatives, which are relatively more stable than their +II state, readily oxidized to their +IV state, and frequently polymerized [61]. Organotin compounds are formed when tin (Sn) bonds with carbon (C) atoms [20,23,62]. A general formula for these compounds is R_x_Sn(L)_4−x_, where R represents an alkyl (such as methyl, ethyl, propyl, or butyl) or aryl (such as phenyl) group and L represents an organic or inorganic ligand [28,63]. Organotin compounds can be categorized based on the number of organic moieties on the Sn atom as mono-substituted, di-substituted, tri-substituted, or tetra-substituted (Figure 1) [28,31,43,64].

The dithiocarbamate anions with the generic formula -S_2_CNR′R″ (Figure 2) are the semi-amides of dithiocarbonic acids and sulfur analogs of carbamates (R_2_NCO_2-_) [65,66,67,68]. Since the 1940s, dithiocarbamates have been used as non-systemic pesticides to treat various fungal diseases in numerous crops and ornamental plants [69]. Moreover, these compounds and their derivatives are active agents in pharmacology, medicine, and biochemistry, and are extensively used in inorganic and organic chemistry. Their stability and interesting electrochemical and optical properties render them a valuable tool in research [70]. Previous studies have reported that the two donor sulfur atoms contribute significantly to the metal-binding capabilities of dithiocarbamate ligands [54,66]. These compounds possess well-known ligands that can firmly and selectively bind to a wide range of metal ions [66,71], making them extremely useful for a variety of medicinal and industrial applications.

The local geometry around the Sn (IV) atom is affected by the chelation mode and coordination number of the dithiocarbamates [65,72]. Additionally, the binding properties of the metal ions determine the structure of the resulting metal complexes [54,73]. Dithiocarbamate can coordinate with the tin atom, relying on the coordination number of the Sn (IV) atom, which ranges from four to seven [65,72]. These compounds are bidentate (two S atoms bonded to the central metal atom or ion), anisobidentate bridging (one S atom bonded to the metal ion and the other S atom bridged to an adjacent molecule, forming dissimilar Sn-S bond distances), or monodentate (one S atom attached to the core of metal atom or ion) (Figure 3) [20,65,66,70,72,74,75,76]. However, as reported by Awang et al., these anions usually behave in a bidentate manner [70].

## 3. Synthesis of Organotin (IV) Dithiocarbamate

The general formula of the organotin (IV) dithiocarbamate is represented as R*_n_*Sn(S_2_CNR′R″)_4−_*_n_* (*n* = 1, 2, or 3), where R, R′, and R″ represent alkyl or aryl groups [77]. Adeyemi and Onwudiwe reported that there is no specific method for preparing organotin (IV) dithiocarbamate complexes; instead, various techniques have been used [20]. Perry and Geanangle documented that metal dithiocarbamate (DTC) complexes are commonly prepared via two synthetic routes: (a) metathetical reactions of alkali-DTC salts with metal compounds and (b) reactions of metal hydroxides with carbon disulfide and the corresponding amines [78]. However, several studies have shown that the in situ method is the most effective way to synthesize organotin (IV) dithiocarbamate compounds [44,53,54,65,77,79,80,81,82,83,84,85,86].

The in situ method has been proposed as the best way to prepare the dithiocarbamate compounds because the ligand cannot be synthesized in a solid form at a temperature above 4 °C [65]. The reaction between carbon disulfide and secondary amines is exothermic (heat release) during the production of dithiocarbamic acid [44,87]. Higher temperatures cause the ligand to break down, resulting in the formation of carbon dioxide, hydrogen sulfide, and ammonium thiocyanide [65]. Domazetis, Magee, and James (1977) prepared triphenyltin (IV) dithiocarbamate compounds using a low-temperature method, resulting in a good yield and high purity of compounds [74]. This shows that the temperature highly influences the form of the product. However, these complexes are very stable at ambient temperatures and begin to melt at temperatures exceeding 100 °C [65].

Previous studies reported that the yield using this method was greater than 50% [44,53,54,80,81,82,85]. Awang et al. and Muthalib and Baba noted in their studies that the dithiocarbamate ligands were synthesized by the nucleophilic addition of carbon disulfide to the corresponding amines in cold ethanol solutions (<4 °C) [77,86]. Generally, the addition of reagents (amines, bases, and carbon disulfide) for the synthesis of dithiocarbamates does not affect the product formed provided that the correct stoichiometric proportions are used [75]. The synthesis of the organotin (IV) dithiocarbamate complex was achieved by adding a defined amount of organotin (IV) chloride dropwise to a stirred mixture of ligands. The white precipitate that developed at the end of the process was filtered, washed with ethanol, and vacuum-dried in a desiccator over silica gel [77,86]. The compounds formed were washed with cold ethanol to remove unwanted residues from the desired product [75]. The narrow melting point intervals of approximately 1–2 °C indicated good compound purity [54]. Figure 1 and Figure 2 show general reactions for complex synthesis with high yields (>70%).

The labile chloride ion is readily displaced by the dithiocarbamate ligand, and the molar ratio of the dithiocarbamate ligand to the organotin (IV) salt varies with the anions (e.g., Cl^−^) present in the organotin (IV) salt [20]. Kamaludin and Awang (2014) reported complexes with the general formula R*_n_*Sn[S_2_CN(C_2_H_5_)(C_6_H_5_)]_4−*n*_ (where R = Bu and Ph for *n* = 2; R = Ph for *n* = 3) derived from the reaction between *N*-ethyl-*N*-phenyldithiocarbamate and the corresponding organotin (IV) salts using the molar ratios of 1:2 for di-organotin (IV) and 1:1 for tri-organotin (IV) compounds [65]. The compounds obtained from the reaction in Figure 3 had a range of yields from 32.0 to 84.5%.

Organotin (IV) dithiocarbamate complexes have also been prepared using a slightly modified procedure. Mohamad, Awang, and Kamaludin (2016), Adeyemi, Onwudiwe, and Hosten (2018), and Adeyemi et al. (2020) synthesized ligands by adding an ammonia solution to an amine solution prior to adding carbon disulfide [53,80,85]. The ammonia solution added to the reaction provided the basic conditions necessary for the reaction to occur [85].

## 4. Characterization of Organotin (IV) Dithiocarbamate

### 4.1. Elemental Analysis of Organotin (IV) Dithiocarbamate (CHNS)

The proportions of carbon, hydrogen, nitrogen, and sulfur in the organic compounds were determined using CHNS elemental analysis. The process of breaking down organic materials results in the release of gases, which can then be analyzed to determine the elements present in the sample [88]. Based on the analysis, when all values are within an adequate range and the experimental elemental proportions match the theoretical values (Table 1), the Sn atom is chelated with a dithiocarbamate group to create neutral compounds [66].

### 4.2. Fourier Transform Infrared Spectroscopy (FTIR) Analysis

Fourier transform infrared (FTIR) is a well-known technique used to identify organic materials, including the structure, behavior, and surroundings of molecules in their native environments [89]. In addition, it is used to determine the coordination modes and binding properties of metal complexes [44]. The FTIR spectrum of the sample is often recorded in the 4000–400 cm^−1^ range [90,91,92,93,94,95]. The presence of a functional group in a molecule or sample that changes its electric dipole moment during vibration is a prerequisite for obtaining an infrared spectrum [90]. There are four important absorption regions in the infrared spectrum, υ(C---N), υ(C-S), υ(Sn-C), and υ(Sn-S), indicating the formation of the organotin (IV) dithiocarbamate complexes discussed in this review (Table 2). 

Dithiocarbamates can be identified by distinct infrared spectral bands corresponding to thioureide ν(C---N) and ν(C-S) bands [44,68,85,86,96,97]. These two stretching frequencies are of particular interest in the IR spectra because they can be used to distinguish between the mono- and bidentate modes of dithiocarbamate ligand binding [98]. In addition, the presence of these two bands confirms that the dithiocarbamate ligands are coordinated to the Sn atom via thiol-sulfur [77]. The thioureide band is a combination of v(C–N) and v(C=N) [85]. This band is a set of carbon–nitrogen bonds that fall between single bonds at 1250–1350 cm^−1^ and double bonds at 1640–1690 cm^−1^ [96,99,100]. However, these bands are often observed in the spectral range of 1450–1550 cm^−1^ [101,102], which is associated with the vibration of the partial double bond and a polar character [20,103].

The partial double-bond character of this thioureide band may be due to the delocalization of electrons in the –NCS_2_ region [54,85,100]. This band is sensitive to the presence of substituent groups on the Sn atoms. The stretching of the (C---N) bond can be explained by the fact that the more electronegative the substituent, the lower the electron density on the Sn atom, and thus the greater the contribution of the canonical structure (Figure 4). This results in stronger double-bond character in the (C---N) bond [104]. 

In addition, more electronegative substituents increase the vibration frequency [44,85,104,105]. Onwudiwe et al. observed an upward shift of approximately 15 cm^−1^ after complexation due to the movement of the electron cloud of the –NCS_2_ group closer to the metal center [97]. This agrees with the results of previous studies that obtained higher stretching frequencies of 1503–1478 cm^−1^ and 1479–1475 cm^−1^ due to the removal of electrons from the tin center of the compound [44,53]. One study found that the stereochemistry of the complex and oxidation state of the metal influenced the frequency in the following order: planar > tetrahedral > octahedral [106]. Awang et al. observed that the C-N bands shifted to lower frequencies owing to an increase in the coordination number [70]. This demonstrates that the geometry of the compound may affect the degree of interaction between the dithiocarbamate ligand and metal ion, resulting in a decreased stretching frequency.

The number of peaks in the υ(C-S) band region can be used to determine the chelation mode of the ligand to the central Sn (IV) atom [54,103]. This υ(C-S) stretching band usually appears in the range of 950–1050 cm^−1^ [65,72,101,102,107]. There are two types of υ (C-S) bands, υ(CS2) asymmetry and υ(CS2) symmetry, which are found at 1120–1131 cm^−1^ and 995–1008 cm^−1^, respectively [98]. The single peak in the spectrum is associated with bidentate symmetrical bonding, whereas the splitting of this band into a doublet may be attributed to the asymmetrical monodentate nature of the dithiocarbamate moiety [108]. Previous studies found a single υ(C-S) band at the absorption peak near 1000 cm^−1^, suggesting a bidentate mode of coordination [44,53,57,77]. Bands split by a difference of 20 cm^−1^ or more within the same region indicate that the ligand coordinates to the central Sn (IV) atom in a monodentate manner [67,102,103].

Organotin (IV) compounds can be identified by the presence of two vibrating bands: υ(Sn-C) and υ(Sn-S). Adeyemi et al. indicated that peaks attributed to the Sn-C stretching frequencies were 512–507 cm^−1^ [53]. These results are consistent with studies by Haezam et al. who found vibrational bands of Sn-C in the range of 546–556 cm^−1^ within the complex, indicating the stretching frequency for compounds with aryl groups [44]. In contrast, Kamaludin and Awang found that Sn-C stretching occurred in diphenyltin (IV) and triphenyltin (IV) compounds with aryl groups at 256 and 261 cm^−1^, and in dibutyltin (IV) compounds with alkyl groups at 554 cm^−1^ [65].

The Sn-S stretching vibration commonly appears in the far-IR region, indicating the presence of metal–ligand bonds and complexation. Muthalib et al. reported a strong absorption band attributed to Sn-S stretching frequencies in the range of 325–386 cm^−1^ [105]. This agrees with previously published data for organotin (IV) dithiocarbamate compounds, with Sn-S stretching vibrations recorded in the regions of 355–377 cm^−1^ and 384–390 cm^−1^ [65,109].

### 4.3. Nuclear Magnetic Resonance (NMR) Spectroscopy

NMR spectroscopy is an advanced technique for the characterization of organotin (IV) complexes. This technique provides useful information for predicting the geometries of organotin (IV) complexes.

#### 4.3.1. ^1^HNMR Spectroscopy

The ^1^H NMR spectrum of the complex can be assigned to two signal regions: methyl and methylene protons in the dithiocarbamate ligand (3–4 ppm), organic groups attached to the Sn atoms of the aliphatic groups (1–2 ppm), and the phenyl group (7–8 ppm) [85]. Adeyemi et al. observed ^1^H NMR chemical shifts of the complexes in the range of δH 3.94–3.56 ppm and δH 4.16–4.11 ppm, attributed to methylene protons directly attached to the N atoms of the dithiocarbamate ligands [53,110].

The alkyl groups present within the complex affect the chemical shifts in the ^1^H NMR spectra of the dithiocarbamate moiety [81]. Onwudiwe and Ajibade [111] observed a shift downfield by δ = 0.4–0.6 ppm for methyl linked directly to N atoms in contrast to the findings of Riveros et al. [112], who observed a chemical shift at δH 3.26–3.40 ppm. A downfield shift may contribute to the electronegativity of the nitrogen atom compared with that of the alkyl carbon [111,113]. In addition, the electronegativity of the coordinated dithiocarbamate group is higher than that of the uncoordinated group [114].

Adeyemi et al. [81] observed that the proton signals of the methyl group of the *N*-ethyl dithiocarbamate appeared to have lower chemical shifts than the *N*-methyl group at 1.23 ppm and 3.80 ppm, likely due to the deshielding effect that was reduced as the distance of the alkyl chain from the thioureide bond or metal center increased [115,116]. This result suggests that the resonances of the methyl protons connected to the N atom are unaffected by the differences in the organotin used to form each complex, as the influence of substituents decreases rapidly with distance [85].

Studies have provided compelling evidence that organic groups attached to nitrogen atoms and Sn (IV) atoms produce distinct proton signals [53,105]. For example [105], the *N*-methyl proton signal was observed as a singlet at 3.19–3.34, whereas multiplet signals at 1.12–1.95 ppm and 4.50–4.65 ppm were assigned to *N*-cyclohexyl protons. For the isopropyl group, multiplet signals of 5.08–5.10 ppm and a doublet of 1.21–1.26 ppm were assigned to methyne and methyl protons, respectively. The chemical shifts of 7.27–8.07 ppm indicated the proton aromatic signals of the phenyl group (C_6_H_5_-N) [54,116].

In addition, Adeyemi et al. [53] reported proton signals in the organotin moiety for the dimethyltin (IV) derivative that appeared in the upfield region as a singlet at 1.53 ppm [77]. For the dibutyltin (IV) derivative, the proton signals were identified in the range of 2.35–0.88 ppm, assigned to methylene and methyl protons of the butyl group [117]. A diphenyltin (IV) derivative was found in the range of 8.06–7.78 ppm, which was consistent with signals reported for aromatic protons bonded to the Sn atom [79,105]. For the phenyl group, the coupling constants for the protons were difficult to measure because the signals appeared as multiplets [65]. The multiplet resonances may be due to the overlapping of proton signals in the aromatic group attributed to the phenyl ring attached to the N atom of the ligand and the phenyl group attached to the central Sn (IV) atom [54].

#### 4.3.2. ^13^C NMR Spectroscopy 

The chemical shift of the CS_2_ peak of the thioureide carbon (–NCS_2_) group is the most significant shift characteristic of the dithiocarbamate complexes. This peak usually occurs between δ 185–220 ppm [118], and the presence of the signal confirms that the coordination of sulfur to the metal atom has occurred [44,70,82]. Onwudiwe and Ajibade (2011) observed weak signals in the ^13^C NMR spectrum for the NCS_2_ carbons of the dithiocarbamate complex at 190.51–202.10 ppm [111], in line with one previous study (196.8–201.9 ppm) [86] and another (197.79–200.82 ppm) [44]. According to Adeyemi, Onwudiwe, and Hosten (2019), the presence of a double bond within the thioureide group is linked to the lower values of the NCS_2_ peaks [81]. Dithiocarbamate ligands with greater trans effects have more shielded -CS_2_ groups, indicating a greater electron density of the ligating group [119].

Furthermore, carbon signals for other carbon-containing substances were observed by ^13^C NMR spectroscopy. Signals attributed to the methylene carbon close to the electronegative nitrogen atom have been observed in the range of δC 55.59–57.05 ppm [84], δC 57.0–59.8 ppm [53], and δC 56.56–57.84 [85]. Recent studies have shown that the methylene C signal has a downfield shift, which can be attributed to the electronegative effect of the N atoms [54]. The electronegativity effects of the N atom cause a downfield shift in the spectrum, which is 20-fold greater for ^13^C than for the ^1^H chemical shift [54,120].

Adeyemi et al. reported that carbon signals in the alkyl substituents attached to the Sn center of a dimethyl complex resonated at δ 15.0 ppm [53], which agreed with previous studies [77]. For the dibutyl complex, the resonation was observed between δ 31.2 and 13.9 ppm [121], whereas the diphenyl complex was found in the region of δ 136.2–128.9 ppm, consistent with the resonation at δ 119.0–135.75 ppm reported by Haezam et al. [44]. Sometimes, two signal sets of aromatic carbon can be observed, indicating that both dithiocarbamate moieties on the adjacent sides of the Sn metal are unequal or magnetically unpaired with the aromatic groups [81,116].

#### 4.3.3. ^119^Sn NMR Spectroscopy

^119^Sn NMR spectroscopy usually determines the coordination number of Sn [105] and provides information on the geometry of organotin (IV) complexes [121]. The ^119^Sn shift depends on the nature of the group attached to Sn. The R substituent of the dithiocarbamate group attached to the Sn atom affects the chemical shift of ^119^Sn, although each compound has the same coordination number [122] due to the sensitivity of the chemical environment of Sn [105,123]. In addition, the nature of the chelating ligand (X) in R_n_SnX_4−n_ can affect the values of −(^119^Sn), with a higher electronegativity of the coordinate ligand causing the −(^119^Sn) value to shift downfield [121,124].

The ^119^Sn NMR spectrum of an organotin (IV) complex often shows a singlet that is significantly lower in frequency than that of the corresponding organotin (IV) salts. The lower chemical shifts of ^119^Sn are mainly due to the presence of electronegative substituents and dπ-pπ bonding effects, which lead to changes in the coordination numbers and bond angles near the Sn center [125]. Hence, a lower chemical shift indicates an increase in the coordination number [122,124]. The ^119^Sn chemical shift value (δ) at −335 ppm (Figure 5) indicates a hexacoordinated geometry around the Sn metal [53]. The list below classifies the most common structures of organotin (IV) compounds according to the coordination numbers [20,126,127,128,129]:Four-coordinate compounds (δ = 200 to −60 ppm): Distorted tetrahedral structuresFive-coordinate compounds (δ = −90 to −190 ppm): Distorted trigonal bipyramidal structureSix-coordinate compounds (δ = −210 to −400 ppm): Distorted octahedral structureSeven-coordinate compounds (δ = −338 to −446 ppm): Distorted pentagonal bipyramidal structure

### 4.4. Recrystallization and Crystallography Study

X-ray crystallography can be used to confirm the structural geometry of the organotin (IV) complexes. Generally, a crystallographic study involves four important steps: crystallization, data collection, structural elucidation, and refinement/verification [130]. The crystallization of organotin (IV) dithiocarbamate complexes is often achieved by dissolving the compound in a mixture of organic solvents, such as chloroform:ethanol, at ratios of 1:1, 2:1, or 1:3. The mixture is allowed to evaporate for a few days at room temperature before being collected and analyzed using X-ray diffraction. Colorless crystals were obtained from previous studies [44,65,109].

Deschamps (2010) suggested a similar technique for the crystallization of a small molecule that was used to produce a single crystal suitable for X-ray diffraction experiments [130]. Figure 6 shows a different type of crystal formation in which some compounds crystallize as thin plates that stick together or form stacks that appear to be single (a), which is unsuitable for X-ray diffraction. However, careful dissection (such as agglomeration) can yield useful single crystals (d). In addition, rapid crystal growth often results in the growth of many crystals from a single nucleation center, forming elongated or acicular crystals (b). They can also be separated to yield useful single crystals (e). Crystal forms resembling glass wool or dust balls (c) are among the least desirable. However, careful dissection can separate a single crystal from one of these undesirable forms (f). Crystals that are as small as 5 × 40 × 220 microns can be used for data collection [130,131].

According to Kim et al., the coordination geometries of organotin (IV) complexes depend on the bonding mode of the dithiocarbamate moiety, which is predominantly monodentate or bidentate [132]. In addition, the overall coordination number at the Sn atom decreases with the number of organic substituents on the Sn atom, which usually occurs when sulfur donors are asymmetrically coordinated [127]. This phenomenon affects the structural geometry of the complex. Organotin (IV) complexes exist in various structures depending on their coordination environments [14,127,128]. However, distorted tetrahedral, trigonal bipyramidal, and octahedral structures are the most common geometries in organotin (IV) dithiocarbamate complexes.

Tri-organotin (IV) complexes are often seen as having a tetrahedral or trigonal bipyramidal symmetry with distortions, regardless of the nature of the ligands attached to the Sn atoms [133]. The distortion arises as a result of the distance of the second uncoordinated sulfur, which forms a “pendant-like” structure [20]. As shown in Figure 7 and Figure 8, the complexes exhibit a distorted tetrahedral geometry around the Sn center [54,65]. The dithiocarbamate ligands are coordinated to the Sn atom in a monodentate manner because of the large disparity in the C–S bond distances. Consistently, the Sn-S bond distances are also non-equivalent for both complexes. The C–S bond angles for complex (C_6_H_5_)_3_Sn[S_2_CN(C_4_H_9_)(C_6_H_5_) are S1–C1 = 1.758(2) Å and S2–C1 = 1.675(2) Å, whereas the (C_6_H_5_)_3_Sn[S_2_CN(C_2_H_5_)(C_6_H_5_) complex with two independent molecules in the asymmetric unit has bond angles of S1-C1 = 1.759(2) Å; S2-C1 = 1.680(2) Å; S3-C28 = 1.7496(19) Å, S4-C28 = 1.6862(19) Å.

The covalent Sn–S bond distances are unequal for both complexes with Sn-S1 (2.4772(5) Å) and Sn-S2 (3.1048(5) Å) for triphenyltin (IV) *N*-butyl-*N*-phenyldithiocarbamate, whereas for triphenyltin (IV) *N*-ethyl-*N*-phenyldithiocarbamate, they are Sn1-S1 = 2.4539(5) Å and Sn1-S2 = 3.1477(6) Å. The longer bond distances (3.1048(5) Å and 3.1477(6) Å) are smaller than the sum of the van der Waals radii of the two atoms (4.0 Å), indicating a weak interaction for the Sn1–S2 bond [20]. The wider angles between the bond distances also cause deviations in the tetrahedral angles from the ideal tetrahedral geometry (109.5°), which is attributed to the influence of the proximity of the non-coordinating thione-sulfur atoms in the complexes [54,134].

The C–N bond length, also known as the thioureide distance, indicates the formation of the dithiocarbamate group. This bond is usually found with a bond length of mean value 1.45 Å [54,135]. However, the thioureide distances in both complexes are shorter than the normal bond length: C1-N1 = 1.337(2) Å for triphenyltin (IV) *N*-butyl-*N*-phenyldithiocarbamate, and C1-N1 = 1.342(2) Å and C28-N2 = 1.333(2) Å for triphenyltin (IV) *N*-ethyl-*N*-phenyldithiocarbamate. Therefore, the bonds formed impose a partial double-bond feature, as observed in most dithiocarbamate complexes [54,65,87,136].

Six-coordinate or octahedral geometries around Sn atoms are the most common for diorganotin (IV) complexes. This geometry is usually formed through the bidentate coordination of the dithiocarbamate ligand with its two sulfur atoms at the Sn center [127]. Awang and Baba (2012) reported the anisobidentate coordination of the dithiocarbamate ligand to the Sn atom through four Sn-S bonds, forming a six-coordinate geometry (Figure 9) [68]. The covalent Sn–S bonds were Sn(1)-S(1) = 2.9255(11) Å; Sn(1)-S(3) = 2.8922(9) Å; Sn(1)-S(2) = 2.5419(10) Å; and Sn(1)-S(4) = 2.5293(10) Å). The longer Sn-S distances are significantly lower compared to the sum of the van der Waals radii (4.0 Å) [137]. Therefore, these bonds can be considered weak [92].

The reported C-S bond distances [S(2)-C(9) = 1.746(3) Å, S(4)-C(18) = 1.743(4)] and [C(9)-S(1) = 1.692(4), C(18)-S(3) = 1.692(4)] of this complex are similar to the dithiocarbamate complexes found by Rehman et al. [138], thus confirming the considerable double-bond character associated with the C-S bonds. The short thioureide distance (C(9)−N(1) = 1.328(4) Å and C(18)−N(2) = 1.331(4) Å) showed that the π-electron density was delocalized over the S_2_CN moiety and had a partial double-bond character [54,65].

Haezam et al. (2021) and Adeyemi, Onwudiwe, and Hosten (2019) reported that the octahedral environment around the tin atom results from the bidentate coordination of sulfur atoms in the ligand dithiocarbamate to the Sn atom in the organotin (IV) complex, as shown in Figure 10 and Figure 11 [44,81]. The sulfur atoms in the structure form unequal distances from the Sn atoms, forming one short and one long Sn-S bond. This suggests that the dithiocarbamate ligand is coordinated asymmetrically with the Sn atom, distorting the regular octahedral geometry. Adeyemi et al. reported that the Sn-S distances in dimethyltin (IV) *N*,*N*-methyl phenyl-*N*,*N*-ethyl phenyl dithiocarbamate were higher than the sum of the usual Sn-S covalent radii (2.42 Å), but significantly less than the sum of the van der Waals radii (3.97 Å) [139], making them a true bond. The distortion around the metal center and the asymmetrical nature of the dithiocarbamate ligand suggest that this complex has a skewed trapezoidal bipyramidal geometry [81].

The crystal structure of 1,1-dibutyl-1,1-bis[(4-methyl-1-piperidinyl)dithiocarbamato)]tin (IV) (Figure 12) in an asymmetric unit with four coordination numbers for the Sn atom was previously reported [92]. The ligand dithiocarbamate bonds to the Sn atom through sulfur atoms (1) and (3), with Sn-S (1) = 2.534(12) Å and Sn-S (3) = 2.536(11) Å. The structural data suggest that the ligand is coordinated in a monodentate fashion, which is unusual for diorganotin (IV) substituents [20]. The Sn-S (2) and Sn-S (4) bond distances, which are 2.918(14) and 2.919(13) Å, respectively, are too long to be strong covalent bonds and are considered weak due to being shorter than the sum of the van der Waals radii for these atoms [109,140].

The bond distance is considered weak, possibly because the steric interaction between the two butyl groups and the four-membered chelating ring prevents the formation of Sn(1)–S(2) and Sn(1)–S(4) bonds [92,109]. The geometry is distorted from a regular tetrahedron because the C (15)-Sn-C (19) bond angle is 135.3° (16°), which is larger than the expected angle for a tetrahedron (109.5°) [20,54]. Another important distortion is caused by the asymmetric Sn-S bond lengths, in which the S(1)–Sn(1)–S(3) angle of 83.03 (14)°, is not consistent with true tetrahedral geometry. Thus, the coordination geometry of this complex can be described as a distorted tetragonal structure.

The monorganotin (IV) complex exhibits only distorted octahedral geometries, with the exception of the pentagonal pyramid geometry [127]. Muthalib and Baba [86] reported the structure of monorganotin (IV) complexes with the formulas of PhSnCl[S_2_CN(Et)(*i-Pr*)]_2_ (compound 5), MeSnCl[S_2_CN(Me)(Cy)]_2_ (compound 11) and MeSnCl[S_2_CN(*i-Pr*)(CH_2_Ph)]_2_ (compound 17) (Figure 13). These molecules have a distorted octahedral geometry, which is attributed to the bonding between the Sn atom and the CClS_4_ donor atom from the two chelating dithiocarbamate ligands. These three compounds have short Sn-S bond distances (from 2.5 to 2.7 Å), which indicate symmetric coordination modes [141]. The shortest Sn-S bond lengths (Sn-S1 = 2.5191(11) Å) observed in compound 5 might be due to the higher electronegative effect of the phenyl substituent.

## 5. Anticancer Effect of Organotin (IV) Dithiocarbamate

The FDA approval of cisplatin (**Pt1**) for the treatment of testicular cancer in 1978 caused a surge in interest in clinical metallodrugs and marked the beginning of medicinal inorganic chemistry [142]. However, cisplatin has significant side effects, including nephrotoxicity, hepatotoxicity, gastrotoxicity, myelosuppression, neurotoxicity, cardiotoxicity, and ototoxicity [10,143]. Therefore, researchers have focused on non-platinum chemotherapy drugs that have fewer side effects [25,36].

Recently, organotin (IV) dithiocarbamate complexes have received considerable attention because of their therapeutic potential. Both organotin and dithiocarbamate moieties have been found to play significant roles in the cytotoxic activities against various cancer cell lines [53]. Organotin compounds have potential as non-platinum chemotherapeutic drugs owing to their ability to exhibit fewer side effects, greater excretion abilities, higher antiproliferative activities, and lower toxicity than other platinum-based drugs [23,37,41,144]. Varela-Ramirez et al. [38] reported that although organotin has been implicated in important deleterious ecological effects, it is possible that by chemical modification, these compounds can be generated with fewer toxic side effects and higher antitumor activity. Therefore, more stable Sn-based compounds with different ligands have been synthesized and tested as potential cancer treatments.

Kamaludin et al. (2013) and Muhammad et al. (2022) claimed that organotin (IV) toxicity was directly correlated with the number and nature of organic moieties [54,145]. Highly substituted organotin compounds are more toxic, whereas shorter alkyl substituents enhance their cytotoxic effects [146,147,148]. In contrast, according to Adeyemi et al. (2020), longer chains of alkyl or aryl groups in organotin complexes cause more cytotoxicity than their shorter-chain counterparts. However, this trend can be a hindrance because of selectivity towards the cell lines used [53]. This could be because the toxicity trend, based on the length and nature of the substituents, depends on the target.

Organotin (IV) compounds with trialkyl and triaryl substituents are more toxic than those with dialkyl and aryl substitutions [35,37,44,55,56,77,79,84,149]. Di-substituted alkyl or aryl Sn(IV) groups showed a better cytotoxic effect compared to mono-substituted derivatives [48,79,107,141,150]. However, monosubstituted alkyl or aryl tin complexes may have good activity, especially butyl and phenyl derivatives [48], possibly because these complexes have exceptional cytotoxic capabilities. Table 3 shows the cytotoxicity of the organotin (IV) dithiocarbamate complexes against various human tumor cell lines (IC_50_ values). Compounds are considered highly toxic if their IC_50_ values are lower than 5.0 μg cm^−3^ (<8.70 μM) [54,151] (Table 4).

The lipophilicity of metal complexes is often affected by the nature of the alkyl or aryl groups on the tin metal core. For example, the phenyl group in an organotin molecule can facilitate π-π interactions with biomolecules [51], contributing to enhanced lipophilicity [48]. Previous research has shown that compounds containing phenyl groups have the highest cytotoxicity activity compared to the other series of complexes being studied, with the lowest IC_50_ value of 0.01 μM [37,48,53,109]. Adeyemi et al. [150] revealed that compounds containing more phenyl substituents possess a higher lipophilicity and thus exhibit a greater cytotoxic effect. This can be attributed to the reduction in the polarity surrounding the Sn metal center, which is ascribed to the decreased atomic dipole moment observed in the central Sn atom of the diphenyltin complex. Consequently, an increase in lipophilicity can enhance the permeability of the complex through cellular membranes.

Additionally, Adeyemi et al. reported that even though the diphenyltin complex is more effective than its mono-phenyl counterpart and conventional medication, it has indiscriminate effects on both healthy and cancer cell lines [150]. This may be attributed to chemical and anatomical barriers hindering the successful delivery of various micro/macromolecular compounds to their intended targets [152]. This issue can be resolved using drug carriers. Drugs carriers, for example nanomaterials, could amplify the potency of pharmaceuticals in cancer treatment. The application of stimuli-responsive systems, such as pH- and photo-responsive biomaterials, has exhibited efficacy in the transportation of drugs to specific target sites [152]. Recently, the green synthesis of nanoparticles using plant sources has also been explored. These biological methods are ecofriendly, consume less energy, and are cost-effective because they do not involve the use of toxic chemicals in their synthesis [153]. Notably, biosynthesized silver-nanoparticle-mediated *Diospyros kaki* L. (persimmon) showed potent cytotoxic effects on the studied cell lines [153]. Nanomaterials have long been utilized because of their remarkable potential to serve as effective carriers for delivering metal-based drugs, owing to their ability to protect active components from degradation, enhance their therapeutic properties, increase drug availability and specificity, and improve solubility [154]. These nanomaterials can release their contents within cells or in the extracellular environment for direct drug absorption and targeted action while preventing unwanted interactions with non-target tissues. When required, they prolong drug circulation and enable sustained drug release [155].

In a previous study conducted by Corvo et al. [156], the encapsulation of a cyclic trinuclear complex of Sn (IV) possessing an aromatic oximehydroxamic acid moiety (MG85) within PEGylated liposomes led to increased cancer cell death in colorectal carcinoma (HCT116) cells compared to the free complex, while concurrently decreasing cytotoxicity to non-tumor cells. Another study conducted by Paredes et al. [154] demonstrated the effectiveness of a nanotheranostic drug, MSN-AP-FA-PEP-S-Sn-AX (AX-3), in targeting and treating a triple-negative breast cancer cell line (MDA-MB-231). By combining receptor-mediated targeting with a specific release mechanism of the organotin metallodrug, AX-3 showed both diagnostic and therapeutic benefits while minimizing the toxic effects on the liver and kidneys upon repeated administration of the multifunctional nanodrug. This indicates that although organotin-based drugs show great promise, their limitations necessitate the use of suitable vectors for biomedical applications.

In addition, the cytotoxicity of the organotin (IV) complex is influenced by the dithiocarbamate ligand. Ligand systems have been reported to play significant roles in the lipophilicity and stability of metal complexes [157]. The presence of sulfur donor atoms in the dithiocarbamate ligand aids the transport of the metal complexes. The chelation effect due to the polarity of the Sn metal enhances biological activities [48] by increasing lipophilicity and facilitating the transportation of molecules to target sites [53,54,59]. Therefore, organotin (IV) compounds can interact with cellular and cytoplasmic membranes [158].

Previous studies have found that the activity of organotin (IV) antitumor compounds is influenced by several factors, including the stability of ligand–Sn bonds (such as Sn-N, Sn-S, and Sn-O), their slow hydrolytic decomposition [159], the structure of the molecule, and the coordination number of the Sn (IV) atoms [160]. Metal complexes formed by bidentate ligands such as dithiocarbamate form relatively stable molecules because of the chelation effect and the fact that the decomposition and loss of the ligand dithiocarbamate are not possible. In addition, the presence of a chelating dithiocarbamate should render the coordination of additional S-donor ligands (e.g., methionine and cysteine residues) trans to the -NCSS moiety less favorable because of the strong trans-influencing effect of the dithiocarbamate sulfur atoms. This may prevent further interactions of the metal center with other thiol-containing biomolecules, which are likely to cause severe side effects, such as nephrotoxicity [161,162]. Kadu et al. found that sulfur-containing compounds had better therapeutic indices in acidic environments; slightly acidic conditions are typically observed in solid tumors induced by the anaerobic fermentation of glucose-secreting lactic acid in tumor tissues [51].

Organotin-induced apoptosis may be the primary mechanism underlying organotin-induced cell death for organotin (IV) complexes [23,37,41,44,55,56,163]. Jakšić [164] proposed that organotins may induce apoptosis by causing changes in the cytoskeleton and disrupting mitochondrial functions. The apoptotic pathway is initiated by the interaction of organotins with cellular components, which can lead to the perturbation of intracellular Ca^2+^ homeostasis and increased [Ca ^2+^] uptake that leads to harmful effects for the mitochondrion, such as loss of mitochondrial membrane potential (ΔΨm), increased ROS production, followed by mitochondrial permeability transition (MPT) and membrane depolarization. The final ΔΨm degradation by MPT promotes the release of cytochrome c from the mitochondria into the cytosol, formation of the apoptosome, and subsequent activation of the initiator caspase-9 and executioner caspase-3, which execute the final steps of apoptosis.

Recent studies have reported the mechanism underlying the anti-proliferative effects of organotin (IV) dithiocarbamate in leukemia cells [55,56], which was supported by the observation of phosphatidylserine exposure on the plasma membrane [55]. Syed Annuar et al. [55] observed that the Ph_3_Sn(*N*,*N*-diisopropyldithiocarbamate) (OC2) complex triggered apoptosis in K562 cells via an intrinsic mitochondrial pathway that was activated by DNA damage, a crucial precursor of apoptosis. Subsequently, OC2 generated an overabundance of reactive oxygen species within the cell. The effect of this oxidative stress was confirmed by a notable decrease in both GSH levels and the percentage of apoptotic cells in the cells pretreated with NAC. Furthermore, research has indicated that organotin (IV) dithiocarbamate compounds can induce cell cycle arrest at different phases, including G0/G1, S, and S-G2/M [55,56]. Cell cycle arrest is crucial for the proper development and survival of multicellular organisms. This event is often triggered in response to the abnormal proliferation or harmful stressors, effectively preventing the spread of dysfunctional cells [165].

The interaction of organotin compounds with biomolecular proteins is influenced by their coordination geometry, biological properties, and presence of functional groups. It has been reported that compounds with a lower coordination number of Sn atoms (i.e., four) are more exposed to interactions with the donor atoms of the target cell biomolecules. Therefore, this complex has a higher anticancer activity [145]. In addition, organotin (IV) compounds exhibit electrophilic properties that enhance their interaction with the electron-donating groups of biomolecules [64], a trait similar to the aqueous form of cisplatin, which is a potent electrophile that reacts with a variety of nucleophiles, including nucleic acids and sulfhydryl groups of proteins [166]. The interaction of this compound with phosphorus-containing biomolecules, such as phospholipids, ATP, and nucleic acids, inhibits the synthesis of phospholipids and the intracellular transport of these biomolecules, thereby inducing the antiproliferative activity of the organotin (IV) derivative complex [64].

Organotin (IV) compounds have been shown to cause DNA damage by binding to the phosphate backbone of DNA, leading to contraction and changes in the DNA conformation [167,168]. The interaction of the organotin complex with DNA differs from that of cisplatin, which can bind to DNA via cross-linking [20]. Studies have shown that intercalation serves as the binding mode of organotin (IV) complexes with DNA [169,170,171]. The DNA-binding capability of organotin compounds is contingent upon factors such as the coordination number, nature of the alkyl groups attached to the central tin atom, and ligands attached to the organotin moiety [169]. According to previous reports, the planar complex exhibits the ability to easily intercalate DNA base pairs in cell lines [172]. Therefore, the complex may be highly cytotoxic. This argument is supported by phenyl-substituted compounds showing higher cytotoxicity than butyl- and methyl-substituted compounds due to the presence of planar phenyl group(s) within the organotin moiety, which enhances the lipophilicity of the complexes and their subsequent penetration into organisms [48]. Furthermore, organotin can inhibit cell division and proliferation by interacting with the nitrogenous bases of nucleic acid nucleotides, interfering with the replication and transcription of DNA molecules or affecting the multienzyme complexes responsible for the replication and transcription of DNA [173]. Hence, these complexes may target DNA, according to previous findings [23,41,48,77,79].

## 6. Conclusions

In this review, we describe the synthesis of organotin (IV) dithiocarbamate complexes using an in situ method. We also highlight the different coordination modes of dithiocarbamates to Sn (IV) atoms: monodentate, bidentate, and anisobidentate. Depending on the coordination mode, the structural geometry of organotin (IV) dithiocarbamate may exist in tetrahedral, trigonal bipyramidal, octahedral, or pentagonal bipyramidal structures. Organotin (IV) constituents play a critical role in inducing cytotoxicity, with ligands involved in transporting the molecule to the target, while avoiding unwanted changes within the biomolecules. The presence of the functional group R on the Sn (IV) atom influences the anticancer activity of these complexes. Therefore, the organotin (IV) dithiocarbamate complex may be a potential anticancer agent; however, more research is needed to understand the mechanism of cell death caused by this complex.

## Data Availability

All data used to support the findings of this study are included in the article.

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
