# Peer review of "Organotin (IV) Dithiocarbamate Compounds as Anticancer Agents: A Review of Syntheses and Cytotoxicity Studies"

_molecules, 2023, doi:10.3390/molecules28155841_

Round 1

Reviewer 1 Report

In the current review, the authors discussed the in-situ approach and various molar ratios used to synthesis organotin(IV) dithiocarbamate complexes. Furthermore, the authors emphasized the several coordination modalities of dithiocarbamates to the tin(IV) atom, including monodentate, bidentate, and anisobidentate. Both the writing and the conclusions are adequately supported by the evidence; the reference list is also quite important.

The following should be taken into account:

·         The introduction should be updated to reflect the significance of advanced studies for this research and the discussion should go into further depth.

·         Most of the figures and tables in the review need to be redrew and redone because they are so poor.

·         Manuscript should be checked for clerical errors. Make sure there are no typographical, spacing, or font writing mistakes in the review.

·         Some references must be formatted in accordance with the authors' instructions.

Make sure there are no typographical, spacing, or font writing mistakes in the review.

Author Response

Dear Reviewer 1,

We would like to express our gratitude for receiving our manuscript. Thank you so much for your suggestion as well. Indeed, your comment helped us in improvising our manuscript. Hence, here we attached our revised manuscript with highlighted changes that we made accordingly. Thank you.

Please find below the author's response. Thank you. 

The following should be taken into account:

  • The introduction should be updated to reflect the significance of advanced studies for this research and the discussion should go into further depth.

Author response: Thank you for your suggestion. We have updated the introduction by adding information related to cancer as well as a metal-based chemotherapy drug in the revised manuscript.

  • Most of the figures and tables in the review need to be redrew and redone because they are so poor.

Author response: Thank you for pointing this out. We have updated the figure by re-drawing and increasing the resolution.

  • The manuscript should be checked for clerical errors. Make sure there are no typographical, spacing, or font writing mistakes in the review.

Author response: Thank you for highlighting the errors. We have updated the revised manuscript accordingly.

  • Some references must be formatted in accordance with the authors' instructions

Author response: Sorry for overlooking. The references have been updated.

Reviewer 2 Report

This manuscript deals with "Organotin(IV) Dithiocarbamate Compounds as an anticancer agent: A review of Synthesis and Cytotoxicity Study" I suggest a minor correction and require a detailed clarification. A correction should be addressed by the authors as follows: The abstract is not well organized; the sentences are incomplete, and there is no sense of continuity. It would be feasible if you included the significance of the current study in the abstract. A brief description of how the authors selected information from the literature in the databases, as well as what time period they searched for, is missing. The authors should justify and expand the information on the advantages of multi-vesicular vesicles for biomedical applications. Authors should specify the main experimental conditions used based on the evidence from the literature. Where they briefly describe the most important data reported in the literature in a homogeneous manner and reinforce the relevance of Organotin(IV) Dithiocarbamate as novel alternatives. Authors should discuss whether the use of Organotin(IV) Dithiocarbamate represents a solid alternative to existing therapeutics. Also, please discuss the use of Organotin(IV) Dithiocarbamate using green nanomaterials to targeting cells and mitochondria . Please add the below studies to your manuscript in the discussion section and bold your study novelties:   -Khalilov, R. A COMPREHENSIVE REVIEW OF ADVANCED NANO-BIOMATERIALS IN REGENERATIVE MEDICINE AND DRUG DELIVERY.Advances in Biology & Earth Sciences Vol.8, No.1, 2023, pp.5-18    -Eftekhari, Aziz, et al. "Natural and synthetic nanovectors for cancer therapy." Nanotheranostics 7.3 (2023): 236.  

Author Response

Dear Reviewer 2,

We would like to express our gratitude for receiving our manuscript. Thank you so much for your suggestion as well. Indeed, the suggested articles helped us in improvising our manuscript. Hence, here we attached our revised manuscript with highlighted changes that we made accordingly.

Please find below the author's response. Thank you.

This manuscript deals with "Organotin(IV) Dithiocarbamate Compounds as an anticancer agent: A review of Synthesis and Cytotoxicity Study" I suggest a minor correction and require a detailed clarification. A correction should be addressed by the authors as follows: The abstract is not well organized; the sentences are incomplete, and there is no sense of continuity.

Author response: Thank you for your suggestion. We have updated the abstract as the revised manuscript.

It would be feasible if you included the significance of the current study in the abstract. A brief description of how the authors selected information from the literature in the databases, as well as what time period they searched for, is missing. 

Author response: Thank you for your comment. The information on the cytotoxicity study of organotin(IV) dithiocarbamates was obtained from article published in 2010 onwards. We would like to include as per your suggestion, however, we have decide to go for mini-review, therefore we not including the method of selecting information including the time period of searching the article.

The authors should justify and expand the information on the advantages of multi-vesicular vesicles for biomedical applications. Authors should specify the main experimental conditions used based on the evidence from the literature. Where they briefly describe the most important data reported in the literature in a homogeneous manner and reinforce the relevance of Organotin(IV) Dithiocarbamate as novel alternative. Authors should discuss whether the use of Organotin(IV) Dithiocarbamate represents a solid alternative to existing therapeutics. Also, please discuss the use of Organotin(IV) Dithiocarbamate using green nanomaterials to targeting cells and mitochondria . Please add the below studies to your manuscript in the discussion section and bold your study novelties:

  -Khalilov, R. A COMPREHENSIVE REVIEW OF ADVANCED NANO-BIOMATERIALS IN REGENERATIVE MEDICINE AND DRUG DELIVERY.Advances in Biology & Earth Sciences Vol.8, No.1, 2023, pp.5-18    -Eftekhari, Aziz, et al. "Natural and synthetic nanovectors for cancer therapy." Nanotheranostics 7.3 (2023): 236

Author response: For the above suggestion, we have found that it is very useful for our study. We have taken this into consideration and added in the discussion. However, the study on nanoparticles of organotin(IV) dithiocarbamates still remains unclear. Therefore, we can’t elaborate specifically on the organotin(IV) dithiocarbamate for this section. Please see below or refer to lines 642 to 677 for the information that we have added to the manuscript.   

“ Additonally, study by by Adeyemi et al., also reported that even though diphenyltin complex exhibits greater effectiveness as compared to its mono-phenyl counterpart and conventional medication, it also has indiscriminate effects on both healthy and cancerous cell lines [151].  This may be attributed to the chemical and anatomical barriers that hinder the successful delivery of various micro/macromolecular compounds to their intended targets [153]. The resolution of this matter can be achieved through the utilization of drug carriers. Drugs carriers by using nano-materials, for instance, could amplify the potency of pharmaceuticals in the treatment of cancer. The application of stimuli-responsive systems, such as pH and photo-responsive biomaterials, has exhibited efficacy in the transportation of drugs to specific target sites [153]. Recently, green synthesis of nanoparticles have also been explored. This biological methods are more eco-friendly, low energy consumption, and cost-effectiveness, as it does not involve the use of toxic chemicals in the synthesis process. Notably, the biosynthesized silver nanoparticles mediated Diospyros kaki L. (Persimmon)had shown potent cytotoxic effect on the cell lines being studied [154]. Generally, nanomaterials have been utilized due to their potential to act as effective carriers for delivering metal-based medication such as ability to protect the active components from degradation, enhance their therapeutic properties, increase drug availability and specificity, or improve solubility [155]. These nanomaterials can release their contents within cells or the extracellular environment for direct drug absorption and targeted action, while preventing unwanted interactions with non-target tissues. When necessary, they can also prolong drug circulation and enable sustained release [156].

In a prior study conducted by Corvo et al., [157] it was discovered that the encapsulation of a cyclic trinuclear complex of Sn(IV) possessing an aromatic oximehydroxamic acid moiety (MG85) within PEGylated liposomes led to increased cancer cell death of colorectal carcinoma (HCT116) cells, when compared to the free complex, while concurrently decreasing the amount of cytotoxicity observed in non-tumor cells [157]. Another study conducted by Paredes and colleagues [155] demonstrated the effectiveness of a nanotheranostic drug called MSN-AP-FA-PEP-S-Sn-AX (AX-3) in targeting and treating triple negative breast cancer cell line (MDA-MB-231). By combining receptor-mediated targeting with a specific release mechanism of organotin metallodrug, the AX-3 drug showed both diagnostic and therapeutic benefits while minimizing toxic effects on the liver and kidneys upon the repeated administration of the multifunctional nanodrug. This indicate that although organotin-based drugs show great promise, their limitations necessitate the use of a suitable vector for their biomedical application. ‘’

Round 2

Reviewer 1 Report

In this revised review, there is some points to take into account, and the review needs to be examined for these mistakes.

·         References shouldn't be included in the conclusion paragraph.

·         Table 1, Table 2 and Figure 5, is inserted between bracts in the text and changed to (Table 1), (Table 2) and (Figure 5).

·         (Figure 4) and (Figures 6-13) are not mentioned in the context.

·         There are still writing faults in both linguistic and scientific writing.

·         Also, some references must be formatted in accordance with the authors' instructions.

As a result, I think the review still need minimal editing in order to reach publication-level status.

Writing errors still exist in both linguistic and scientific writing.

·  

Author Response

Dear Reviewer 1,

Thank you for your comment and suggestion about our manuscript. Please see the attachment.  Thank you.
